# Nitrogen Signaling Genes and *SOC1* Determine the Flowering Time in a Reciprocal Negative Feedback Loop in Chinese Cabbage (*Brassica rapa* L.) Based on CRISPR/Cas9-Mediated Mutagenesis of Multiple *BrSOC1* Homologs

**DOI:** 10.3390/ijms22094631

**Published:** 2021-04-28

**Authors:** Haemyeong Jung, Areum Lee, Seung Hee Jo, Hyun Ji Park, Won Yong Jung, Hyun-Soon Kim, Hyo-Jun Lee, Seon-Geum Jeong, Youn-Sung Kim, Hye Sun Cho

**Affiliations:** 1Plant Systems Engineering Research Center, Korea Research Institute of Bioscience and Biotechnology, Daejeon 34141, Korea; hmjung@kribb.re.kr (H.J.); lar1027@kribb.re.kr (A.L.); chohee0720@kribb.re.kr (S.H.J.); hjpark@kribb.re.kr (H.J.P.); jwy95@kribb.re.kr (W.Y.J.); hyuns@kribb.re.kr (H.-S.K.); hyojunlee@kribb.re.kr (H.-J.L.); 2Department of Biosystems and Bioengineering, KRIBB School of Biotechnology, Korea University of Science and Technology (UST), Daejeon 34113, Korea; 3Department of Functional Genomics, KRIBB School of Biotechnology, Korea University of Science and Technology (UST), Daejeon 34113, Korea; 4Department of Biotechnology, NongWoo Bio, Anseong 17558, Korea; sgjeong09@nongwoobio.co.kr

**Keywords:** Chinese cabbage (*Brassica rapa* L.), flowering time, CRISPR/Cas9, nitric oxide signaling, *BrSOC1*, *BrNIA1*, *BrNIR1*, RNA-seq, vernalization, late bolting

## Abstract

Precise flowering timing is critical for the plant life cycle. Here, we examined the molecular mechanisms and regulatory network associated with flowering in Chinese cabbage (*Brassica rapa* L.) by comparative transcriptome profiling of two Chinese cabbage inbred lines, “4004” (early bolting) and “50” (late bolting). RNA-Seq and quantitative reverse transcription PCR (qPCR) analyses showed that two positive nitric oxide (NO) signaling regulator genes, *nitrite reductase* (*BrNIR*) and *nitrate reductase* (*BrNIA*), were up-regulated in line “50” with or without vernalization. In agreement with the transcription analysis, the shoots in line “50” had substantially higher nitrogen levels than those in “4004”. Upon vernalization, the flowering repressor gene *Circadian 1* (*BrCIR1*) was significantly up-regulated in line “50”, whereas the flowering enhancer genes named *SUPPRESSOR OF OVEREXPRESSION OF CONSTANCE 1* homologs (*BrSOC1s*) were substantially up-regulated in line “4004”. CRISPR/Cas9-mediated mutagenesis in Chinese cabbage demonstrated that the *BrSOC1-1/1-2/1-3* genes were involved in late flowering, and their expression was mutually exclusive with that of the nitrogen signaling genes. Thus, we identified two flowering mechanisms in Chinese cabbage: a reciprocal negative feedback loop between nitrogen signaling genes (*BrNIA1* and *BrNIR1*) and *BrSOC1s* to control flowering time and positive feedback control of the expression of *BrSOC1s*.

## 1. Introduction

Flowering is a central event in the plant life cycle. Flowering time plasticity has evolved in relation to seasonal and developmental changes to maximize reproductive success in various environments [1]. It has emerged as one of the key traits affecting crop yield in commercial agronomic and horticultural crops because biotic and abiotic stresses can be avoided by modifying the flowering time [2,3]. Flowering is generally influenced by various hormones and nutrients. Nitric oxide (NO) is a central growth regulator known to inhibit flowering. In Arabidopsis, NO represses the expression of flowering activator genes CONSTANS (CO) and GIGANTEA (GI), which function in the photoperiod pathway, and promotes the expression of flowering repressor gene *FLOWERING LOCUS C* (*FLC*) [4]. Furthermore, enzymes involved in NO signaling, nitrite reductase (NIR) and nitrate reductase (NIA), are associated with flowering. The *nia1 nia2* double knockout mutant of Arabidopsis shows faster bolting than the wild type (WT). Moreover, the expression of *NIA1* is antagonistic to that of the flowering integrator gene *SUPPRESSOR OF OVEREXPRESSION OF CONSTANCE 1* (*SOC1*), depending on the NO concentration [5]. Recently, transcriptome profiling showed that the photoperiod pathway genes are potentially involved in nitrogen (N)-dependent flowering in Arabidopsis [6]. Moreover, nitrogen response in flowering had been reported in crops [7,8]. Thus, N is believed to cause late flowering, and research on the relationship between flowering time genes and N metabolism is currently underway.

*Brassica* is a key agricultural genus of the Brassicaceae family, which includes a diversity of health-promoting vegetable crops, such as cabbage, Chinese cabbage, and broccoli, along with oilseed crops, such as canola and mustard [9]. Leafy vegetable crop varieties of *Brassica rapa* and *Brassica oleracea* require precisely regulated flowering time to achieve optimal crop productivity. Early bolting in *Brassica* species severely decreases the commercial value of the crop, while delayed flowering interferes with seed production [10]. *B. rapa* and *B. oleracea* plants exhibit a strong requirement for vernalization, i.e., long-term exposure to low temperature for floral induction [11,12,13]. Although *B. rapa* and *B. oleracea* belong to the same genus, they exhibit different responses to vernalization; *B. rapa*, like *Arabidopsis thaliana*, shows seed-responsive vernalization, whereas *B. oleracea* exhibits plant-responsive vernalization [14,15]. Unlike annual plants, which do not require vernalization, biennial species depend on vernalization to transition from the vegetative to the reproductive phase because of allelic variation at two genes, *FRIGIDA* (*FRI*) and *FLC* [16]. A thorough understanding of the regulatory mechanism underlying flowering time in *Brassica* species is a prerequisite to avoiding early bolting. Recently, the genome sequences of *B. rapa* [17], *B. oleracea* [18], and *Brassica nigra* [19] have been published. Additionally, flowering time genes were recently identified in the genus *Brassica* by performing comprehensive RNA-Seq analyses of genes differentially expressed under different environmental conditions [12,20,21].

Chinese cabbage (*Brassica rapa* L. ssp. *pekinensis*) is a commercially important vegetable crop consumed worldwide, especially in East Asia. As a seed-vernalizable plant, Chinese cabbage is prone to premature bolting, which affects crop yield and quality, particularly in spring varieties that are usually exposed to a long cold period during winter [22,23,24]. To establish effective molecular markers linked to bolting resistance, numerous quantitative trait loci (QTLs) controlling bolting and flowering time have been identified in Chinese cabbage over the last two decades [25,26,27,28,29,30]. Additionally, a genomic variation map of Chinese cabbage, based on whole-genome resequencing data of 194 geographically diverse accessions, revealed that allelic variation at *BrVIN3.1* and *BrFLC1* represents a major genetic resource between spring Chinese cabbage and autumn Chinese cabbage, ensuring seasonal bolting time [31]. Fine mapping of QTLs is limited by the genetic background, size of the population, and the number of available genetic markers. Therefore, there is a need to identify flowering time loci related to phenotypic differences in bolting time and involved in the molecular mechanism and regulatory network of the flowering pathway [32]. However, despite the genetic dissection of floral transition regulation and identification of flowering time genes in Chinese cabbage, few studies have reported on the flowering mechanism in Chinese cabbage [33,34]. Several recent attempts have been made to introduce the flowering time trait directly by CRISPR technology-mediated genetic variation using DNA-free genome editing [35,36].

We recently identified 223 flowering time genes in Chinese cabbage, of which 50 were expressed in response to vernalization [12]. In this study, we focused on the molecular mechanism of flowering in Chinese cabbage by performing RNA-Seq analysis of two different bolting time inbred lines (“4004” and “50” both are 2*n* = 20) before and after exposure to vernalization conditions. Further, we used the CRISPR system for targeted mutagenesis of *BrSOC1* homologs (*BrSOC1s*) to introduce the late-flowering trait into an elite variety “20” (2*n* = 20). CRISPR/Cas9-mediated multi-gene mutagenesis of *BrSOC1s* gave new insights into the flowering mechanism and demonstrated that genome editing could be applied to improve crop yield in Chinese cabbage.

## 2. Results

### 2.1. Differences in Vernalization-Induced Bolting Time between Lines “4004” and “50”

Our previous work showed that bolting in the inbred line “4004” is induced by vernalization [12]. In this study, we compared the bolting time of two Chinese cabbage inbred lines, “4004” (early bolting) and “50” (late bolting). Seven-day-old seedlings of both lines were vernalized for 0 or 35 days, and plants were grown under long-day conditions for 60 days in soil before assessing the bolting phenotype. Under vernalization conditions, the inbred line “4004” bolted earlier than line “50” (Figure 1a). Consistent with the flowering phenotypes of the two lines, the average time to bolting was 17 days in “4004”, which was approximately 13 days shorter than in line “50”, exhibiting an average time to bolting of 30 days. Additionally, there were fewer rosette leaves in line “4004” than in line “50” (average of the number of leaves at bolting: 9.5 in line “4004” vs. 14.5 in line “50”) (Figure 1b). However, in the absence of vernalization, both inbred lines showed no bolting for 12 weeks (Appendix A). Thus, our results indicated that vernalization has a significant effect on bolting in Chinese cabbage, and plants of inbred lines “4004” and “50” grown from vernalized seedlings displayed distinct phenotypes.

### 2.2. Transcriptome Profiling

To investigate the molecular mechanisms determining the bolting time in Chinese cabbage, we performed comparative RNA-Seq analyses of lines “4004” and “50” vernalized for 0 (0D Ver) and 35 days (35D Ver) (Appendix A). RNA isolates were prepared from the shoots of both inbred lines, and cDNA libraries were sequenced on the Illumina HiSeq 2000 platform in triplicate (primer sets used in this study, Appendix A). The base quality of raw reads was evaluated using a series of pre-processing steps, and 56–76% of the raw reads were retained (Appendix A), indicating the use of relatively stringent pre-processing analysis criteria. Then, clean reads were mapped using the *B. rapa* reference genome (http://brassicadb.org/brad/, accessed on 27 April 2021, version 1.5) (Appendix A). Approximately 70–84% of the clean reads mapped to the coding sequences of genes, indicating that the transcriptome sequences were of good quality. Although pre-processing eliminated a large number of reads, the final mapped reads were suitable for further functional transcript analysis.

### 2.3. Identification of Differentially Expressed Genes (DEGs) and Functional Pathway Enrichment Analysis Comparing Lines “4004” and “50”

The DEGs were identified based on the following criteria: normalized read count ≥ 500, log_2_(fold-change) ≥ 1, and adjusted *p*-value (FDR) < 0.05. The comparison between the two inbred lines (“4004” vs. “50”) identified a total of 1710 DEGs, most of which (1229, 70%) were specific to the non-vernalized (0D Ver) conditions, whereas a less significant proportion of 10% (179) was detected under vernalization conditions (35D Ver) (Figure 2a); 302 DEGs (17.5%) were found under both 0D Ver and 35D Ver conditions. This suggested that the heterogeneity between the two inbred lines based on the up- and down-regulation of DEGs was mostly inherent and not much affected by vernalization.

To assess the physiological role of DEGs, we conducted a gene ontology (GO) enrichment analysis using the GO database (http://www.geneontology.org, accessed on 27 April 2021). A total of 22 significantly enriched pathways were identified under 0D Ver conditions based on an adjusted *p*-value < 0.05 for “4004” vs. “50”: 11 pathways were up-regulated in line “4004” relative to the expression status in line “50”, and the other 11 pathways were up-regulated in line “50” relative to line “4004” (Figure 2b). Under 35D Ver conditions, there were 9 up-regulated pathways in line “4004” and 8 up-regulated pathways in line “50” relative to the expression status in the other inbred line (Figure 2c). Surprisingly, “Nitrogen compound metabolic process” (GO: 0006807) under 0D Ver conditions and “Nitrate reductase activity (GO: 0008940) under 35D Ver conditions had the highest enrichment scores in line “50”: 21.64 and 6.76, respectively, based on -log10 (adjusted *p*-value) scores (Appendix A). Interestingly, nitrogen is known to directly affect the onset of flowering, and in our analysis, the nitrogen-related pathways (keyword: nitrogen, nitric acid, nitrate, and nitrite) were enriched in line “50” under both 0D and 35D Ver conditions. Further analysis of the flowering-related pathways under 0D and 35D Ver conditions gave a consistent result, showing that phenological flowering DEGs were up-regulated in line “4004” under 35D Ver conditions. Specifically, “Positive regulation of flower development” (GO: 0009911) and “Maintenance of inflorescence meristem identity” (GO: 0010077) were up-regulated under 35D Ver in “4004” (Figure 2d, left). Furthermore, “Inflorescence development” (GO: 0010229) and “Inflorescence morphogenesis” (GO: 0048281) were up-regulated under 0D Ver in “4004”, but “Floral organ development” (GO: 0048437) and “Floral whorl development” (GO: 0048438) were up-regulated under 0D Ver in line “50” (Appendix A, left), and “Regulation of meristem development” (GO: 0048509) was up-regulated under 35D Ver in “50” (Appendix A, right). Next, we assessed Br flowering time (Ft) genes from the GO pathways by comparing “4004” with “50”. As shown in Figure 2e, the flowering key regulator homologs *BrSOC1s* (*BrSOC1-1*; Bra004928, *BrSOC1-2*; Bra000393 and *BrSOC1-3*; Bra039324), *BrCRY2* (Bra015313), along with *BrPRR5* (Bra036517), were enriched in nine GO pathways as up-regulated DEGs in “4004” under 35D Ver. Additionally, *BrGID1C* (Bra009970), *BrVRN1* (Bra001729), *BrRAV1* (Bra019821), *BrCRY2s* (Bra015313 and Bra030568) and *BrTEM1* (Bra011002) were enriched in 11 GO pathways as up-regulated DEGs in “4004” under 0D Ver (Appendix A, right). In contrast, Br Ft genes were not detected in “50” (Appendix A). These results showed that Br Ft genes from flowering-related pathways were only detected as up-regulated DEGs in “4004” under 35D Ver conditions, which matched the flowering properties presented in Figure 1a.

### 2.4. Identification of B. rapa Nitrate Reductase Genes and Relative Expression Levels Derived by Comparing Lines “4004” and “50”

Among the DEGs that were up-regulated in line “50” relative to their expression in line “4004”, there were several genes that were highly associated with *“Nitrogen metabolism”* pathways. Therefore, we decided to examine DEGs involved in the NO signaling pathway. To study the functional relevance of nitrogen for flowering in Chinese cabbage, we applied the Arabidopsis research results to Brassicaceae plants. Previous studies had demonstrated that *NIA* and *NIR* were associated with flowering in Arabidopsis. Specifically, the *nia1 nia2* double knockout mutant of Arabidopsis displayed an early-flowering phenotype and antagonistic gene expression compared with *SOC1,* indicating a negative regulation mechanism for flowering [5,37]. Based on this information, we created a schematic diagram showing the negative regulation of flowering by nitrogen (Figure 3a) and assumed that the genes participating in N metabolism, *NIA* and *NIR*, may play an important role in controlling bolting time in lines “4004” and “50”. We examined whether the DEGs *BrNIA* and *BrNIR* that we identified in Chinese cabbage are homologs of Arabidopsis genes by performing a phylogenetic tree analysis based on the predicted amino acid sequences using tblastx. Arabidopsis proteins AtNIA1 (At1g77760) and AtNIR1 (At2g15620) were highly homologous with the corresponding *B. rapa* proteins (Figure 3b). Specifically, *BrNIA* and *BrNIR* genes were identified as DEGs by comparing the two lines with and without vernalization (Appendix A).

To confirm the reliability of RNA-Seq data, the expression levels of two genes, *BrNIA1* (Bra015656) and *BrNIR1* (Bra015227) (shown in the phylogenetic tree), were analyzed by qPCR (Figure 3c). The results of RNA-Seq and qPCR analyses were generally consistent. Overall, Bra015227 (*BrNIR1*) and Bra015656 (*BrNIA1*) had higher expression levels in line “50” than in line “4004” regardless of the vernalization status. Moreover, vernalization (35D Ver) increased the *BrNIR1* expression level by a 2-fold change in line “50” relative to the level in line “4004”. In general, the expression levels of NO signaling genes were well correlated with the bolting phenotypes of lines “4004” and “50”. This suggested that the effect of nitrate signaling on the flowering pathway in Chinese cabbage was similar to that in Arabidopsis.

Then, we examined the nitrate content in lines “4004” and “50” based on the differences in the NO signaling gene expression as previously described [38]. The nitrogen content was substantially higher in line “50” than in line “4004” measured at the 0 and 35 days vernalization time points with a more than 10-fold difference (Figure 3d). This result was consistent with the previous finding that nitrogen starvation rapidly suppressed the NO signaling genes *NIA* (Os02g0770800, Os08g468100) and *NIR* (Os01g0357100) in rice [39]. Our findings supported the notion that *BrNIA* and *BrNIR* were directly involved in nitrate/nitrite assimilation in Chinese cabbage. Thus, we concluded that nitrogen signaling genes were related to the flowering pathway in Chinese cabbage as in the Arabidopsis system.

### 2.5. Identification and Analysis of Flowering Time Genes Differentially Expressed between the Two Inbred Lines

Next, we analyzed the DEGs enriched in the flowering pathway. The DEGs analysis (read count > 500, log_2_(fold-change) ≥ 1, and adjusted *p*-value (FDR) < 0.05) for genes up-regulated in “4004” vs. “50” detected only 12 flowering time-related DEGs (0D Ver: 6 DEGs, and 35D Ver: 6 DEGs) (Appendix A). Interestingly, the three *BrSOC1s* genes related to flowering time expression control were identified as up-regulated flowering time DEGs in line “4004” under 35D Ver conditions, and they also had the tendency that their expression levels were inversely correlated to those of *BrNIA1* and *BrNIR1* (Appendix A). However, it was difficult to broadly identify flowering time-related DEGs based on just 12 DEGs. We reasoned that we could increase the number of flowering time DEGs by adjusting the expression levels via normalized read counts because flowering time genes were expressed at relatively low levels. We observed the pattern of increasing the number of flowering time-related DEGs while lowering the expression level conditions. There were 34 and 15 flowering time-related DEGs without and with vernalization, respectively, using the following criteria: read count > 50, log_2_(fold-change) ≥ 1, and adjusted *p*-value (FDR) < 0.05 (Figure 4a). To assess the molecular function of flowering time DEGs with known roles in the flowering pathway, we examined the ratio of flowering enhancer and repressor genes expressed in both inbred lines with the read count ≥ 50 (Figure 4b). Under 0D Ver conditions, the numbers of up-regulated flowering enhancers vs. repressors among flowering time-related DEGs were similar (16 vs. 17, respectively) in the comparison of “4004” vs. “50” using up-regulating conditions (left graph). Under 35D Ver conditions, there were more up-regulated enhancer genes than repressor genes (9 vs. 2), and the repressor pattern also coincided with their bolting characteristics (1 vs. 3) (right graph). Furthermore, we identified the flowering time DEGs according to the distribution of flowering pathways. Under 0D Ver conditions, the Circadian/Light/Photoperiod (C/L/P) pathway accounted for more than 50% of up-regulated Ft DEGs in the comparison of “4004” vs. “50” (Figure 4c, left graph). In contrast, under 35D Ver conditions, the DEGs belonging to the C/L/P pathway were selectively decreased (Figure 4c, right graph). These results indicated that vernalization diminished the number of up-regulated flowering enhancers in the comparison of “4004” vs. “50” for flowering time DEGs. This applied especially to the flowering time DEGs of the C/L/P pathway, which were more affected by vernalization than those of other flowering pathways.

### 2.6. Validation of the Expression of Flowering Time Genes Involved in the Flowering Regulatory Network

To validate the RNA-Seq data of DEGs involved in the flowering pathway, we conducted a qPCR analysis of flowering time genes exhibiting a 2-fold difference in expression levels between lines “4004” and “50” before and after vernalization (Appendix A, * marked) using the same RNA samples that were used for the RNA-Seq analysis (Figure 5). Most C/L/P pathway genes were induced by vernalization in both inbred lines, except *BrTEM1* (a flowering repressor). The expression levels of C/L/P pathway flowering time genes, *BrCIR1* and *BrTEM1*, were highly correlated between the RNA-Seq and qPCR analyses, although the differences for the *BrCRY2* and *BrPRR5* expression levels between the two inbred lines were smaller with the qPCR method than with the RNA-Seq method. Overall, the expression profile of *BrCIR1* in the two inbred lines, but not that of *BrTEM1*, was consistent with the bolting phenotypes specific for the inbred lines. Next, we examined the expression of *BrVRN1*, which was the only flowering time DEG in the V pathway. The expression level of *BrVRN1* varied substantially between lines “4004” and “50” before vernalization, which was consistent with their respective bolting phenotypes. Moreover, the results of qPCR and RNA-Seq analyses of *BrVRN1* were similar, although there was no significant difference after vernalization. Interestingly, among the flowering time DEGs belonging to the gibberellin and Integrator pathways, the *BrSOC1* homologs (*BrSOC1-1, BrSOC1-2*, and *BrSOC1-3*) reached remarkably high expression levels in line “4004” upon vernalization; however, in line “50”, all *BrSOC1* genes were barely expressed, regardless of vernalization. The *BrGID1C* gene was differentially expressed between the two inbred lines without vernalization but showed no significant difference in expression under vernalization conditions, which was consistent with the RNA-Seq data. The expression of *BrGID1C* in both inbred lines was also consistent with their respective bolting phenotypes, especially under non-vernalization conditions.

We also tested the expression levels of other Ft genes from both inbred lines (Appendix A). The genes of a major flowering determinant, *BrFLC1* and *BrFLC3*, were substantially repressed in both inbred lines after vernalization. Expression of most Ft genes (*BrCCA1, BrCOL1-2, BrCOL9, BrVIN3A, BrGA1, BrTOC1,* and *BrLHY*) was induced by vernalization regardless of the inbred line. The *BrSPA3* and *BrGI* genes were not affected by vernalization. We hypothesized that the *BrSOC1s*, which displayed a distinct difference between “4004” and “50” after vernalization, might have a more important role in regulating the flowering of Chinese cabbage than other Ft genes.

### 2.7. CRISPR/Cas9-Mediated Mutagenesis of Multiple BrSOC1 Homologs Introduced the Late-Bolting Trait into the Elite Line “20”

Based on RNA-seq and qPCR analyses, we identified *BrSOC1s* as the most important genes in vernalization. Then, we employed the CRISPR/Cas9 system to create *BrSOC1s* knockouts, which we used to examine whether these homologs were critical for the flowering phenotype in response to vernalization in Chinese cabbage. For this experiment, we choose the elite line “20” with a bolting time that was typical for *BrSOC1s* knockouts. In addition, we attempted to edit multiple *BrSOC1* genes of (three homologs: *BrSOC1-1/BrSOC1-2/BrSOC1-3*) at once by locating the protospacer adjacent motif (PAM) sequence and designing single-guide RNA (SgRNA) for the 1st exon of the three *BrSOC1s* (Figure 6a). Then, we performed the *BrSOC1s* editing using the Chinese cabbage transformation and regeneration method in the inbred line “20” as previously reported [40]. After co-cultivation of Chinese cabbage and *Agrobacterium tumefaciens* carrying the CRISPR/Cas9 vector system (No. 1 in Figure 6b), we confirmed the shoot and root regeneration in a selection medium (No. 2 and 3 in Figure 6b). Finally, we generated *BrSOC1s*-edited T_0_ plants (No. 4 in Figure 6b). We conducted next-generation sequencing (NGS) of T_0_ plant samples, which confirmed *BrSOC1s*-specific gene editing (Appendix A) based on the insertion/deletion (indel) proportion. The genotype proportion of *BrSOC1-1* (Bra004928) with +1 bp insertion was 93.7% and that of *BrSOC1-2* (Bra000393) with −1 bp deletion and +1 bp insertion combined was 87.4%. In contrast, the *BrSOC1-3* (Bra039324) genotype proportion with +1 bp insertion was 34.0% (Figure 6c). In addition, “20” CRISPR/Cas9-*brsoc1-1/1-2/1-3* T_1_ (B36-1) plants did not display a bolting phenotype until 60 days after the 35 days of vernalization, unlike the “20” WT (Figure 6d). The average time to bolting was approximately 17 days in vernalized WT line “20” plants, but no bolting phenotype was observed within 100 days in the CRISPR/Cas9-mediated *BrSOC1s* mutagenesis T_1_ plants after the vernalization. In addition, the average number of leaves at the time of bolting was approximately 15 in WT line “20”, whereas the leaves could not be enumerated in the genome-edited *brsoc1s* because of leaf senescence, which already occurred during the long growth period (>60 days after 35 days of vernalization) (Figure 6e). Finally, we found that the high proportion of gene-edited *BrSOC1-1* and *BrSOC1-2* was sufficient to introduce the late-bolting trait in Chinese cabbage.

### 2.8. Backcross Breeding for Brsoc1/2/3 Mutagenesis Controlling the Late-Bolting Trait

We harvested 20 seeds from the T_0_ plant and performed NGS analysis, along with flowering phenotype observation in T_1_ plants obtained from 20 seeds. We determined that T_1_ plants, which were typically edited with *BrSOC1-1* and *BrSOC1-2* as in B36-1-9 (*brsoc1-1/1-2*) and *BrSOC1-1*, *BrSOC1-2*, and *BrSOC1-3* as in B36-1-18 (*brsoc1-1/1-2/1-3*), displayed late bolting (Figure 7a). However, T_1_ plants were difficult to use in the experiment because the Cas9 gene was integrated all plants. We then analyzed T_2_ generations of the two T_1_ plants. In addition, Chinese cabbage lines were selected that had the Cas9 gene removed (Appendix A).

To validate that the CRISPR/Cas9-mediated mutagenesis of *brsoc1s* was linked to the late-bolting trait, the B36-1-9 line was backcrossed (B36-1-9 BC). The indel frequency of *BrSOC1s* was restored to about 50% in B36-1-9 BC T_1_ plants compared to before being B36-1-9 based on the NGS analysis (Appendix A). The B36-1-9 BC T_1_ plants were self-crossed to obtain T_2_ seeds, and 85 seeds of BC36-1-9 BC T_2_ seeds were analyzed for their genotypes and bolting characteristics (Appendix A, Appendix A). Among B36-1-9 BC T_2_ plants, the *brsoc1-1*/*brsoc1-2* plants displayed no bolting phenotype for 40 days after 35 days of vernalization, like B36-1-9 T_2_ plants (Appendix A). Finally, when *BrSOC1-1* or *BrSOC1-2* were backcrossed, the flowering phenotype was restored, suggesting that the *brsoc1-1/brsoc1-2* knockout plants carried the controlled late-flowering trait in Chinese cabbage.

### 2.9. Loss-of-Function BrSOC1s Showed Reciprocal Regulation with Nitrogen Signaling Genes and Nitrogen Content and Preserved Positive Feedback Regulation of the Original BrSOC1s Genes

To assess the molecular characteristics of the genome-edited *brsoc1s* mutants, we performed a qPCR analysis of *BrSOC1s* and nitrogen signaling genes and determined the nitrogen content in T_2_ plants based on the previous results comparing the two inbred lines, “4004” and “50”. Firstly, we determined the expression levels of original *BrSOC1* genes based on the observation that the positive feedback regulation is closely connected to flowering time in Arabidopsis [41]. The WT and B36-1 plants displayed different expression levels and patterns of the *BrSOC1s*. Under 0D Ver, when the *SOC1* expression remained at a basal level, there was no significant difference in the *BrSOC1s* expression levels of the genome-edited plants compared to those in the WT, although *BrSOC1-2* was decreased in the B36-1s. Upon vernalization, the expression levels of all three *BrSOC1s* were markedly induced in the WT line “20” with up to 30-fold change, whereas the B36-1 genome-edited plants did not respond to vernalization, displaying significantly lower expression levels (Figure 7b, upper graphs). The loss-of-function *NIA1* in Arabidopsis had antagonistically increased expression levels of *SOC1*. Therefore, we examined the nitrogen signaling genes *BrNIA1* and *BrNIR1* in B36-1s plants. Remarkably, the expression levels of these genes differed from those in the WT. Without vernalization, the expression level of *BrNIR1* was lower in the B36-1s mutants than in the WT, but there was an approximately 4-fold (B36-1#9) or 10-fold (B36-1#18) expression level increase in response to vernalization. Similarly, the expression level of another nitrogen signaling gene, *BrNIA1,* was up to 40-fold increased after vernalization in B36-1s mutants, compared to the expression level without vernalization (Figure 7b, lower graphs). The significant expression level variations in the nitrogen signaling genes in B36-1s plants led us to the analysis of the nitrogen content. Interestingly, the nitrate content was higher in the B36-1#18 mutant than in the WT (4-fold higher in the 0D Ver sample and 9-fold in the 35D sample). The nitrate content in WT “20” plants was reduced in response to vernalization with a 2-fold change, whereas the B36-1#18 plants displayed no significant variation (Figure 7c). In addition, transcript levels of *BrCIR1* and *BrCRY2* were also decreased in the B36-1s than that of WT “20” under 35D Ver conditions (Appendix A). Although the nitrate content did not completely match the tendency of the nitrogen reductase expression level observed by comparing WT and B36-1s, the data suggested that *BrSOC1s* played a central role in flowering and reversely modulated the expression levels of nitrogen signaling genes in Chinese cabbage. Additionally, this result indicated a *BrSOC1s*-specific control mechanism by positive feedback, which appeared to be conserved between Chinese cabbage and Arabidopsis.

## 3. Discussion

Although the knowledge of the flowering mechanism is of fundamental importance for improving crop productivity, the molecular basis of flowering time control has not yet been elucidated in Chinese cabbage. In this study, we accessed the molecular variation in flowering time between early- and late-bolting inbred lines of Chinese cabbage via RNA-Seq and CRISPR/Cas9 technologies.

We previously detected 223 flowering time genes in the early-bolting Chinese cabbage inbred line “4004”, of which approximately 20% were regulated by vernalization [12]. In the present study, we performed transcriptome profiling of two inbred lines, “4004” and “50”, exhibiting different bolting times. However, comparative transcriptome profiling of these two lines was difficult because of two reasons. Firstly, most of the DEGs identified between the two lines were detected in the absence of vernalization, and only a few DEGs were identified after vernalization (Figure 2). Thus, the DEGs identified between the two lines before vernalization were responsive to vernalization. For instance, the expression of *BrFLCs* was highly down-regulated in both lines after vernalization, despite the significant difference between the two lines prior to vernalization, and the expression of *BrTEM1, BrGID1C*, and *BrVRN1* was also substantially decreased after vernalization in line “4004” but not in line “50” (Appendix A). Secondly, because of the low read counts in our analysis, there was a substantial number of flowering time genes that failed to meet the criteria to be categorized as DEGs in response to the two vernalization conditions or based on the differences between the two inbred lines. Finally, only 5% of the flowering time genes (12 out of 223) were differentially expressed between line “4004” and line “50”. This suggested the possibility that other regulatory pathway genes controlled the bolting time in these two inbred lines.

The identification of DEGs involved in N metabolism is an interesting finding because nitrate or N in plant tissues affects the flowering time [42,43]. Therefore, *BrNIR* and *BrNIA* are good candidates for determining the flowering mechanism in Chinese cabbage. The RNA-Seq and qPCR analyses revealed that the expression levels of *BrNIR* and *BrNIA* were up-regulated in line “50” grown with or without vernalization (Figure 3C). NIR is the main enzyme responsible for the production of NO and affects a wide range of physiological and developmental processes in plants, including flowering timing in Arabidopsis [5,44,45]. In Arabidopsis, flowering is facilitated when plant growth occurs under low-nitrate conditions compared with growth under high-nitrate conditions. Furthermore, the expression of flowering time genes is regulated by nitrate availability; for example, low nitrate conditions repress flowering repressors and activate flowering enhancers [43,46]. Because nitrate availability affects the flowering pathway and the timing of the vegetative-to-reproductive phase transition, we surmise that increased expression of *BrNIR* and *BrNIA* in line “50” contributes to flowering repression in Chinese cabbage, consistent with the study in Arabidopsis [5]. The differentially expressed flowering time genes that affect the N metabolism may comprise a potential regulatory network linked to N signaling and the flowering pathway in Chinese cabbage, as suggested by the findings in Arabidopsis. Our data suggest that *BrCIR1* and *BrSOC1s* are responsive to vernalization (Figure 5). The MYB-related protein CIR1, also known as REV2, is involved in circadian regulation and acts as a negative regulator of flowering in Arabidopsis [47]. Furthermore, nitrate treatment significantly induces *CIR1* expression [6]. This led us to speculate that CIR1 is a critical factor for the N-mediated regulation of flowering time. Surprisingly, *BrCIR1* expression was higher in line “50” than in line “4004” after vernalization (Figure 5). Expression of the floral integrator gene *SOC1* also increases under low-nitrate conditions, leading to accelerated flowering [46]. The difference in the expression level of *BrSOC1s* between the two inbred lines was also correlated with the transcript levels of N metabolism genes *BrNIR* and *BrNIA* (Figure 3 and Figure 5).

CRISPR/Cas9-mediated mutagenesis is a reliable system employed in functional studies on multiple-gene clusters of unknown functions and is becoming one of the most effective tools to create desirable phenotypes in crops relying on transformation technology. Therefore, it is the preferable tool for verifying that the regulation and function of *BrSOC1s* are associated with N signaling in the Chinese cabbage flowering pathway. There are three *SOC1* genes in Chinese cabbage, but it is not known whether all of them are involved in flowering time control. However, their involvement in flowering is considered to be highly likely because their response to vernalization appears to be robust (Figure 5), and the double and triple mutants have a delayed flowering phenotype (Figure 6). Moreover, the knockout mutants of *SOC1* that were backcrossed with WT line “20” appear to indicate that *BrSOC1* and *BrSOC2* are concurrently involved in controlling flowering time in Chinese cabbage (Appendix A).

In the *BrSOC1* triple mutant, the expression of the *BrNIR1* and *BrNIA1* genes was increased after vernalization (Figure 7b), suggesting the possibility that the *SOC1* gene inhibited NO signaling during vernalization treatment, as previously shown in Arabidopsis [5]. It can be presumed that the NO signal and SOC1 form a regulation loop that inhibits each other (Figure 8). In addition, SOC1 expression was not increased in the *brsoc1s* triple mutant even after vernalization. This may occur if the *brsoc1s* mutant transcript is unstable, but it appears to be less likely because it was similar or slightly lower before the vernalization process. Therefore, it is possible that *BrSOC1* expression is regulated by positive feedback after vernalization. These two regulation mechanisms—suppression of negative regulators (NO signal) and positive feedback of itself—can quickly induce the *SOC1* gene expression and strongly increase it after vernalization. More detailed research will be needed to confirm our model. Thus, our results suggest that NO signaling genes play an essential role in the regulation of flowering time in Chinese cabbage by differentially regulating the expression of the key flowering enhancer *SOC1* (Figure 8).

In summary, comparative transcriptome profiling of two inbred lines identified genes with a potentially critical role in regulating the bolting time in Chinese cabbage. Additionally, our data suggest that genes involved in nitrate metabolism also function in the flowering pathway in Chinese cabbage.

## 4. Materials and Methods

### 4.1. Plant Materials and Growth Conditions

Inbred lines of Chinese cabbage (*Brassica rapa* ssp. *pekinensis*), “4004” (early-bolting line), “50” (late-bolting line), and “20” (early-bolting line, elite variety) were used in this study. Seeds were obtained from NongHyup Seed in Korea (Gyeonggi-do, Anseong, Korea). Seeds were sown in sterilized soil and placed in a growth room maintained at 23 °C under long-day conditions (16 h light/8 h dark). After 1 week, vernalization of seeds was initiated in a cold room at 5 ± 1 °C with a 12 h light/12 h dark photoperiod for 35 days. Seeds not subjected to vernalization served as a control. After vernalization, plants were transferred to the growth room and grown for 40 days. The flowering phenotype was assessed based on the rosette leaf numbers and the days to bolting, which were recorded when the length of the floral axis was ≥0.5 cm. All experiments were performed in three independent biological replicates (*n* = 10 per replicate). To conduct RNA-seq analysis, shoot samples from five independent plants without vernalization (0 days) and with vernalization (35 days) were collected at the same time point in the light/dark cycle; samples were collected in three independent biological replicates (Appendix A).

### 4.2. Total RNA Isolation, Library Construction, and RNA-Seq Analysis

Total RNA was extracted from 250 mg shoot tissue of lines “4004” and “50” subjected to vernalization for 0 days (D0_4004 and D0_50) or 35 days (D35_4004 and D35_50) in three biological replicates using RNAiso Plus TRIzol (TaKaRa, Shiga, Japan). For line “4004”, we integrated duplicate RNA-seq data that we previously generated; therefore, only one set of samples was newly analyzed in this line. The purified total RNA was analyzed with a NanoDrop 1000 spectrophotometer (Thermo Fisher, Waltham, MA, USA). Then, 6 μg RNA was used for cDNA library construction after passing an RNA quality evaluation. Subsequently, cDNA sequencing libraries were constructed using the TruSeq RNA Library Prep Kit (Illumina, Inc., San Diego, CA, USA) and sequenced on the Illumina HiSeq™2000 platform (Illumina, Inc., San Diego, CA, USA) to generate 101 bp paired-end reads. All raw sequences were deposited in the GenBank Gene Expression Omnibus (GEO) under the accession numbers GSE106444 (for previous “4004” RNA-seq duplicates) and GSE139375.

### 4.3. Transcriptome Assembly and Mapping

Raw sequence reads were analyzed using the RNA-seq parameter of the Illumina pipeline. The raw reads were filtered using several trimming steps to remove adaptor contamination, low-quality sequences, and reads containing stretches of Ns. The DynamicTrim and LengthSort programs of SolexaQA (v. 1.13) were used on reads with a Phred quality score of 31 (Q ≥ 20) and a minimum length of 25 bp [48] (Appendix A).

The filtered datasets were pooled and mapped using the *Brassica rapa* reference gene set (http://brassicadb.org/brad/, accessed on 27 April 2021, version 1.5). Mapping was analyzed using the Bowtie2 (v2.1.0) program with default settings (mismatch ≤ 2 bp) [49] (Appendix A).

### 4.4. Identification and Analysis of DEGs

Gene expression data of lines “4004” and “50” obtained in this study (a total of eight samples) and of four samples of line “4004” obtained previously [12] (a total of 12 samples) were used for the identification of DEGs. To identify DEGs between two vernalization treatments and between the two inbred lines, raw reads were normalized using the DESeq library in R (v3.5.1) [50]. Genes with normalized read count ≥ 500, log_2_(fold-change) ≥ 1, and adjusted *p*-value (FDR) < 0.05 were designated as differentially expressed. Furthermore, flowering time-related DEGs were analyzed with read count > 50 for the purpose of gathering the number of DEGs (with the same criteria: log_2_(fold-change) ≥ 1, and adjusted *p*-value (FDR) < 0.05). The expression of DEGs in lines “4004” and “50” (4004 vs. 50) was normalized relative to their expression in line “50” treated with vernalization for 0 days (D0_50) to identify up- and down-regulated genes, which were depicted as Venn diagrams drawn using Microsoft PowerPoint and as bar graphs generated using Microsoft Excel. Flowering enhancer/repressor and pathway were analyzed as described previously [12].

### 4.5. Functional Annotation of DEGs

To conduct functional annotation, the annotated genes were compared with gene sequences in the Phytozome database (http://www.phytozome.net/, accessed on 27 April 2021) using BlastP (E values ≤ 1 × 10^−10^) (BLAST v.2.2.28+). A gene ontology (GO) database (http://www.geneontology.org, accessed on 27 April 2021) was used for GO enrichment analysis, and transcripts were annotated using BlastX (E values ≤ 1 × 10^−10^). GO pathway enrichment analysis was conducted using the adjusted *p*-value < 0.05.

### 4.6. Real-Time Reverse Transcription PCR (qPCR) Analysis

To validate the results of RNA-Seq analysis, selected genes were analyzed by qPCR in three biological replicates. Total RNA was treated with DNaseI (Thermo Fisher Scientific, Waltham, MA, USA) to remove genomic DNA contamination, and 2 µg total RNA was used for first-strand cDNA synthesis using PrimeScript^TM^ RT Master Mix (TaKaRa, Shiga, Japan), according to the manufacturer’s instructions. The cDNA was resuspended in nuclease-free water and used for qPCR analysis with a CFX Connect™ Real-Time PCR Detection System (Bio-Rad, Hercules, CA, USA). *BrActin* (Bra022356) was used as an internal control for normalizing mRNA levels. All primer sets used in this study are listed in Appendix A.

### 4.7. Bioinformatics Analysis

Amino acid sequences of AtNIA1 (At1g77760) and AtNIR1 (At2g15620) proteins were used for phylogenetic tree analysis. BrNIA1s and BrNIRs amino acid sequences were constructed using tblastx from transcript id of RNA sequencing data. The phylogenetic tree of NIA1 and NIR1 between Arabidopsis and Chinese cabbage was constructed using the neighbor-joining method in Molecular Evolutionary Genetic Analysis (MEGA; version 7) for whole protein sequences.

### 4.8. Nitrate Assay

The nitrate assay protocol was based on the salicylic acid method [51,52]. The assays were performed using three independent *B. rapa* leaves in the same position as previously described in an *Arabidopsis* methods [38]. Weighed leave samples (up to 100 mg) in a 1.5 mL tube were frozen in liquid nitrogen and milled into powder using a grinder. Then, 1 mL of sterilized deionized water was added into the tube, followed by boiling for 25 min. The samples were centrifuged at 13,000 rpm for 10 min at 4 °C, and 0.1 mL supernatant was transferred into a 15 mL conical tube (Hyundai Micro, Seoul, Korea). Next, 0.4 mL of 5% salicylic acid-sulfate acid solution (5 g salicylic acid (Sigma-Aldrich, St. Louis, MO, USA) in 100 mL sulfate acid (Junsei, Tokyo, Japan) was added, the sample was mixed thoroughly by inverting, the reaction incubation was conducted at 23 °C for 20 min and 9.5 mL of 8% NaOH solution was added. Before measurement, standard curves were checked using KNO_3_^−^ (Sigma-Aldrich, St. Louis, MO, USA; 10 to 120 mg/L concentration), and regression analysis was conducted based on the standard curve. The OD_410_ value was determined after cooling the tube to room temperature; 0.1 mL of sterilized deionized water was used as a control instead of the supernatant.

### 4.9. CRISPR/Cas9-Mediated Mutagenesis of BrSOC1-1/1-2/1-3 and Genetic Transformation

Cas-Designer (http://www.rgenome.net/cas-designer/, accessed on 27 April 2021) was used to design sgRNA. The sequences of the Chinese cabbage *SOC1* genes (Bra000393, Bra004928, and Bra039324) were compared to find a region with matching sequences. Then, sgRNAs were designed (5′-TCTGATCATCTTCTCTCCTAAG-3′) in the corresponding region to target the three *BrSOC1* genes with one sgRNA. The sgRNA was synthesized with a restriction enzyme sequence for cloning, and it was cloned into the pHAtC vector [53] treated with the restriction enzyme AarI (CACCTGC (4/8)^). The recombined vector was transformed into *Agrobacterium tumefaciens* LBA4404 and then used for transformation of Chinese cabbage plants.

The Chinese cabbage elite line “20” (NongHyup Seed. Gyeonggi-do, Anseong Korea) was used for the transformation of Chinese cabbage according to the previously published method [40]. For transformation, hypocotyl was incubated for ~1–2 days in darkness, and then it was cut into segments with a length of 0.5 to 1 cm. Co-culture was performed with the transformed *Agrobacterium tumefaciens* in the dark for 2 days. After washing, the explants were cultivated in a callus induction medium (MS salts, including 3% sucrose, 5 mg/L benzyl adenine (BA), 1 mg/L naphthaleneacetic acid (NAA) and 300 mg/L cefotaxim) in the dark for 3 days. The induced calli were transferred to a shoot induction medium (MS salt, including 3% sucrose, 10 mg/L BA, 1 mg/L cefotaxime, and 10 mg/L hygromycin). Once the shoot was obtained, it was cultured in a root-inducing medium (MS salt, including 3% sucrose, 0.1 mg/L NAA, and 0.1 mg/L gibberellin).

### 4.10. Backcross Breeding for Brsoc1s

All organs, except the pistil, were removed from the “20” buds that were not opened; then, the *Brsoc1/2/3* (B36-1-9, T_1_) pollen was used for crossing. The obtained seeds were sown, and genomic DNA was extracted from each plant. The indel analysis was performed using NGS to confirm that the *Brsoc1/2/3* mutation was hetero. Several confirmed plants were selected, and selfing was performed to obtain seeds of the next generation.

### 4.11. Characterization of Mutations Inherited from the T0 to T2 and the Backcrossing Generations in Transgenic Plants

To examine the mutations of *BrSOC1* genes in transformed Chinese cabbage plants, sequences adjacent to the target positions were amplified with primer sets (*BrSOC1-1*, 5′-AGGTCGTTTTATGTGTATGAC-3′, 5′-ACTCACTTTACAGTTAACAC-3′; *BrSOC1-2*, 5′-CCCAAAGGAAGATTGTATAA-3′, 5′-ACCCTCCCTCTAAGCAAACG-3′; *BrSOC1-3*, 5′-GTTCTCTGCAAGTTAAAAAG-3′, 5′-TGTATAAACCAAGACATACT-3′). The PCR products were subjected to deep sequencing performed at ToolGen Inc. (ToolGen, Seoul, Korea) to detect the mutations.

## Figures and Tables

**Figure 1 ijms-22-04631-f001:**
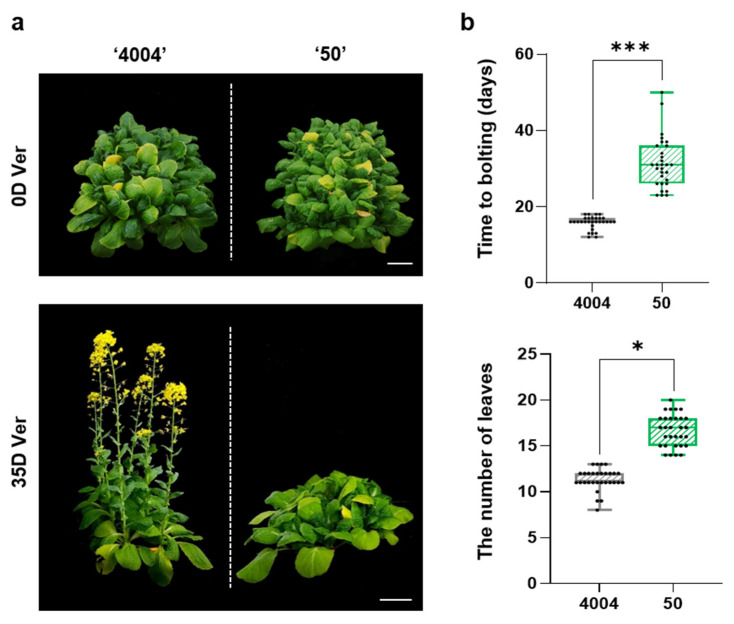
Extensive bolting variation in two Chinese cabbage inbred lines, “4004” and “50”, in response to vernalization. (**a**) Bolting phenotypes of the early-bolting line “4004” and the late-bolting line “50” following vernalization for 0 or 35 days. Non-vernalized seedlings (0 days, 0D Ver) were grown at 23 °C for 30 days. Vernalized seedlings (35 days, 35D Ver) were grown at 5 ± 1 °C and 12 h light/12 h dark photoperiod for 35 days and then transferred to a growth room at 23 °C for 20 days. Scale bar = 5 cm. (**b**) Statistical analysis of the days to bolting and the number of leaves in the two inbred lines. The leaves were counted when the plants started bolting. All experiments were performed in three independent biological replicates (*n* = 10 per replicate). Statistically significant differences are indicated with asterisks (* *p* < 0.05, *** *p* < 0.001; Student’s *t*-test).

**Figure 2 ijms-22-04631-f002:**
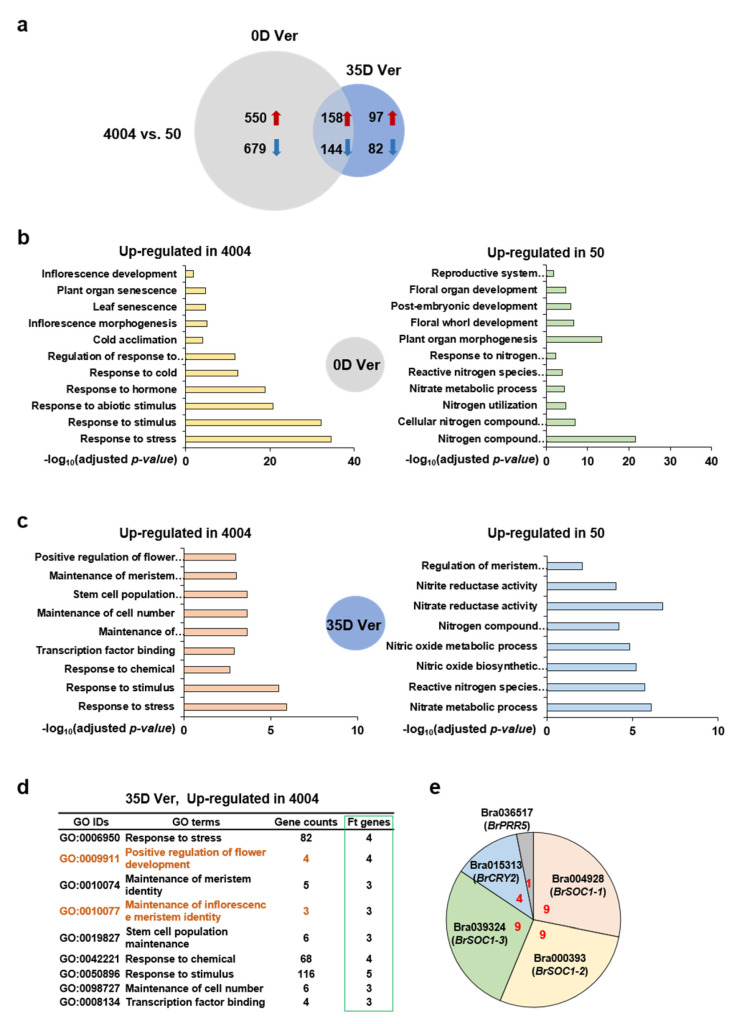
Bioinformatic analysis of RNA-seq data comparing lines “4004” and “50” grown with or without vernalization. (**a**) Differentially expressed genes (DEGs) were detected by comparing “4004” and “50” at each vernalization time point (0D Ver, 0 days vernalization; 35D Ver, 35 days with vernalization). Red arrowheads indicate up-regulated DEGs; blue arrowheads indicate down-regulated DEGs. (**b**,**c**) Gene ontology (GO) enrichment analysis of up- or down-regulated DEGs in “4004” vs. “50” (up-regulated DEGs in “50”) under 0D Ver conditions (**b**) and 35D Ver conditions (**c**). The most enriched GO terms are shown (adjusted *p*-value < 0.05). (**d**) Identification of flowering time genes derived from GO terms up-regulated in “4004” under 35D. The orange font represented GO trems of flowering pathway-related genes. (**e**) Circular graph distribution of flowering time (Ft) genes in Figure 2d. Red number means the number of GO pathways with Ft genes.

**Figure 3 ijms-22-04631-f003:**
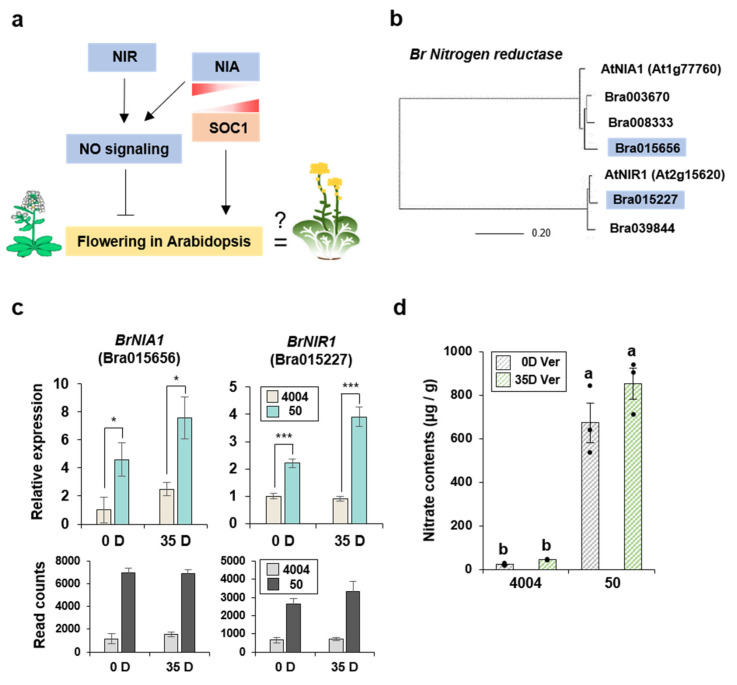
Identification and gene expression levels of nitrogen reductase genes in two inbred lines. (**a**) Schematic flowchart of Arabidopsis studies assessing nitrogen reductase genes and flowering. (**b**) Phylogenetic relationship between Arabidopsis and Chinese cabbage nitrogen reductase-encoding genes. The phylogenetic tree was created using the neighbor-joining method. The scale bar shows evolutionary distances based on amino acid substitutions. (**c**) Quantitative RT-PCR (qPCR) results of nitrogen reductase genes (Bra015656 and Bra015227) compared to that of RNA-seq results. The actin (*BrACT2)* was used as an internal control. In the qPCR analysis, the expression level of each from “4004” on day 0 was defined as “1”. Error bars represent standard error (SE) of three replicates. Statistically significant differences are indicated with asterisks (* *p* < 0.05, *** *p* < 0.001; Student’s *t*-test). (**d**) Nitrogen content in lines “4004” and “50” at each vernalization time point (0 and 35 days). Three black dots indicate each biological replicate (*n* = 3), bar graphs mark the average of biological triplicates, error bars mean ± SE of biological triplicates, and the different letters indicate a significant difference between samples (*p* < 0.05, one-way ANOVA followed by Tukey post-hoc test).

**Figure 4 ijms-22-04631-f004:**
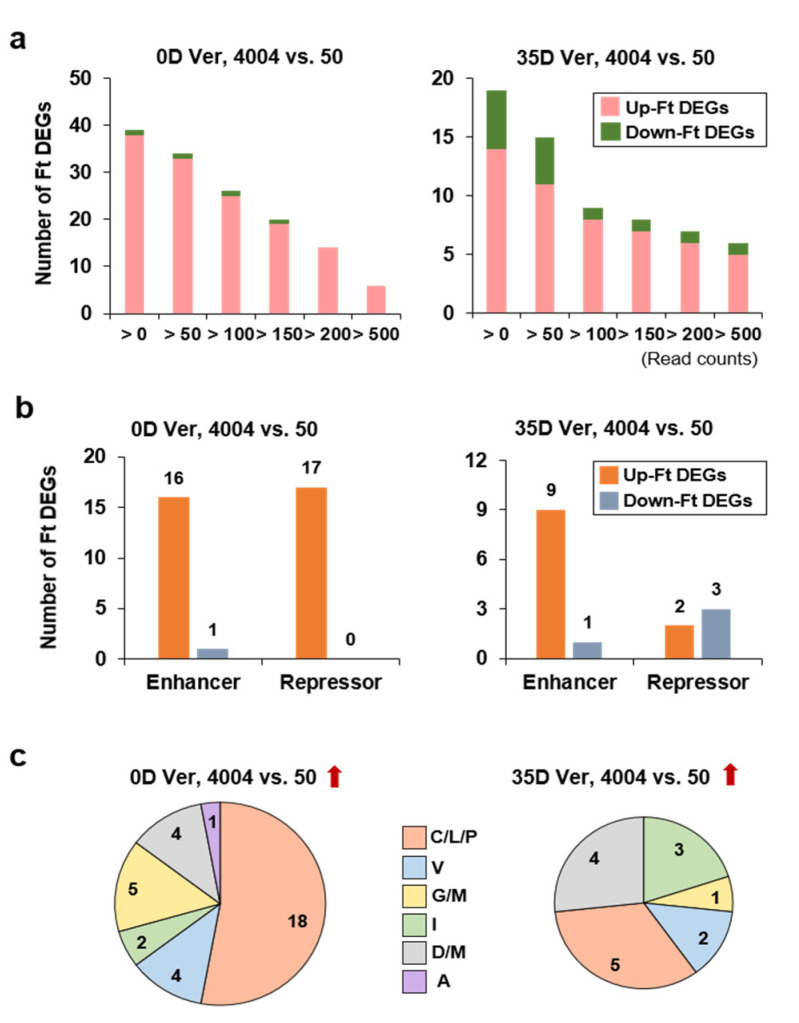
Flowering time-related DEGs based on the comparison between lines “4004” and “50”. (**a**) The expression map of the up-regulated flowering time DEGs according to the expression level comparison of “4004” vs. “50”. The bar graph shows the number of flowering time DEGs based on “4004” vs. “50” plants. (**b**) Classification of flowering time DEGs according to flowering enhancer/repressor criteria. The bar graph shows the number of up-regulated flowering time DEGs based on “4004” vs. “50” plants (read count > 50). (**c**) Circle graph distribution of flowering time DEGs according to flowering pathways. Red arrowheads indicate up-regulated DEGs detected by comparing “4004” vs. “50”. C/L/P, circadian clock, light signaling and photoperiod); V, vernalization; G/M, gibberellin signaling and metabolism; I, integrator; D/M, development and meristem response; A, autonomous.

**Figure 5 ijms-22-04631-f005:**
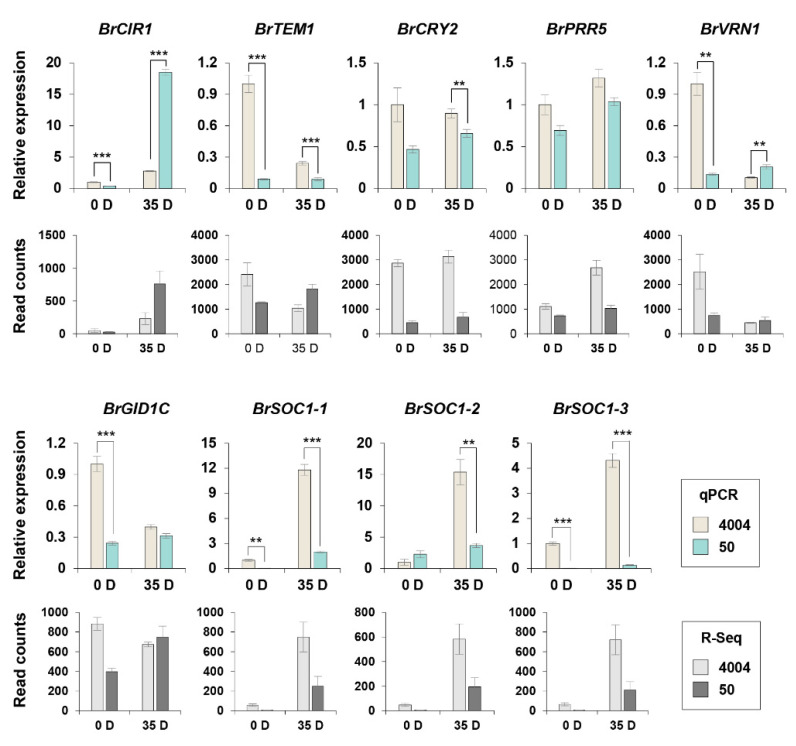
Quantitative RT-PCR (qPCR) analysis of flowering time genes in the two inbred lines cultured with or without vernalization. This analysis was done in the same leaf samples (+/− Vernalization) used for RNA sequencing. The qPCR and RNA-seq results of the target genes were compared to confirm the gene expression effects in relation to the flowering mechanism. The list of target flowering time genes is shown in Appendix A. The expression levels of flowering time DEGs and genes were normalized to actin (*BrACT2*) as a reference gene. The expression levels of genes in line “4004” on day 0 (no vernalization) were defined as “1”. Data points represent mean ± SE of three replicates. 0 D, 0 days vernalization; 35 D, 35 days vernalization. Statistically significant differences are indicated with asterisks (** *p* < 0.05, *** *p* < 0.01; Student’s *t*-test).

**Figure 6 ijms-22-04631-f006:**
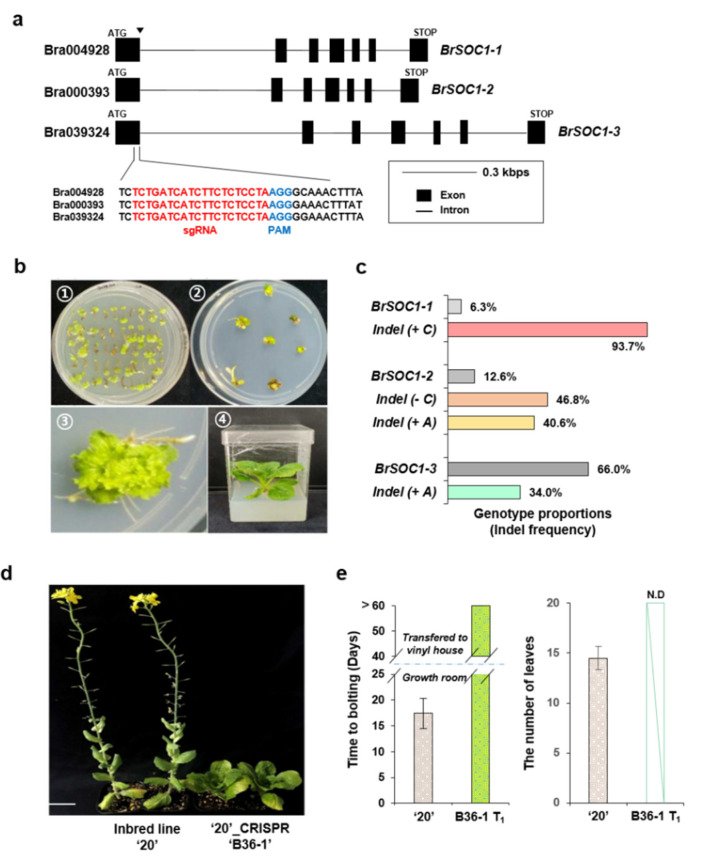
Analysis of *BrSOC1-*encoding genes edited using CRISPR/Cas9 in *Agrobacterium tumefaciens*-mediated Chinese cabbage transformation and regeneration. (**a**) Generation of single-guide RNA (sgRNA) from exon 1 of *BrSOC1s* genomic sequence. (**b**) Chinese cabbage transformation and regeneration process. The pictures show the following: ① culture in a selection medium after co-cultivation of Chinese cabbage and *Agrobacterium tumefaciens*; ② shoot regeneration; ③ magnified picture of shoot regeneration; and ④ plant with root regeneration. (**c**) Analysis of T_0_ plant by next-generation sequencing (NGS). Percentage indicates proportion between wild-type line “20” and insertion/deletion (indel) variants (#) of *BrSOC1s* genome-edited plants. (**d**) Representative plants illustrating the bolting phenotype of wild-type (“20” inbred line with early bolting) and genome-edited T_0_ plants (B36-1 with late bolting). Scale bars = 5 cm. (**e**) Statistical analysis of the days to bolting and number of leaves in “20” and B36-1 under culture room condition for 30 days and then transferred to the vinyl house for over 30 days (*n* = 20, T_0_ generation, error bars = S.D.). N.D, not determined due to late bolting under vinyl house conditions.

**Figure 7 ijms-22-04631-f007:**
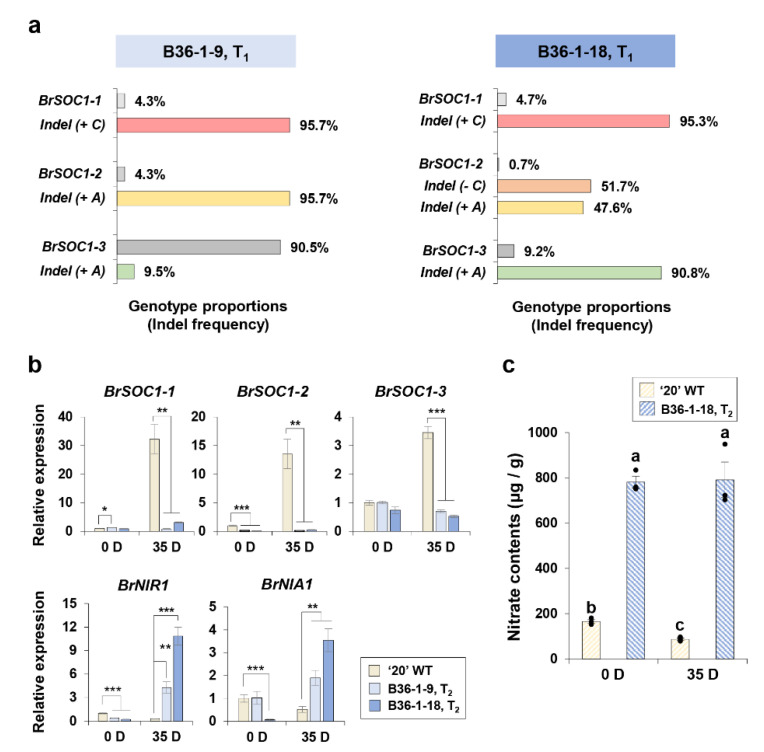
Molecular characterization of CRISPR/Cas9-mediated *BrSOC1s* genome-edited T_2_ plants. (**a**) NGS analysis of T_1_ plant having differential *BrSOC1s* gene editing. Percentage means proportion between WT and indel variants. (**b**) The transcript levels of *BrSOC1s* and nitrogen signaling genes *BrNIR1* and *BrNIA1* in genome-edited B36-1 plants. *BrACT2* was used as an internal control to quantify the transcripts. The expression level of a gene in the “20” wild type (WT) on day 0 (no vernalization) was defined as “1”. Values are mean ± SD. Statistically significant differences are indicated with asterisks (* *p* < 0.05, ** *p* < 0.01, *** *p* < 0.001; Student’s *t*-test). (**c**) Nitrogen content in “20” WT and B36-1 T_2_ #18 at both vernalization time points (0 and 35 days). Three black dots indicate each biological replicates (*n* = 3), the graph bars represent the averages of biological triplicates, error bars mean ± SE of biological triplicates, and different letters indicate a significant difference between samples (*p* < 0.05, one-way ANOVA followed by Tukey post-hoc test).

**Figure 8 ijms-22-04631-f008:**
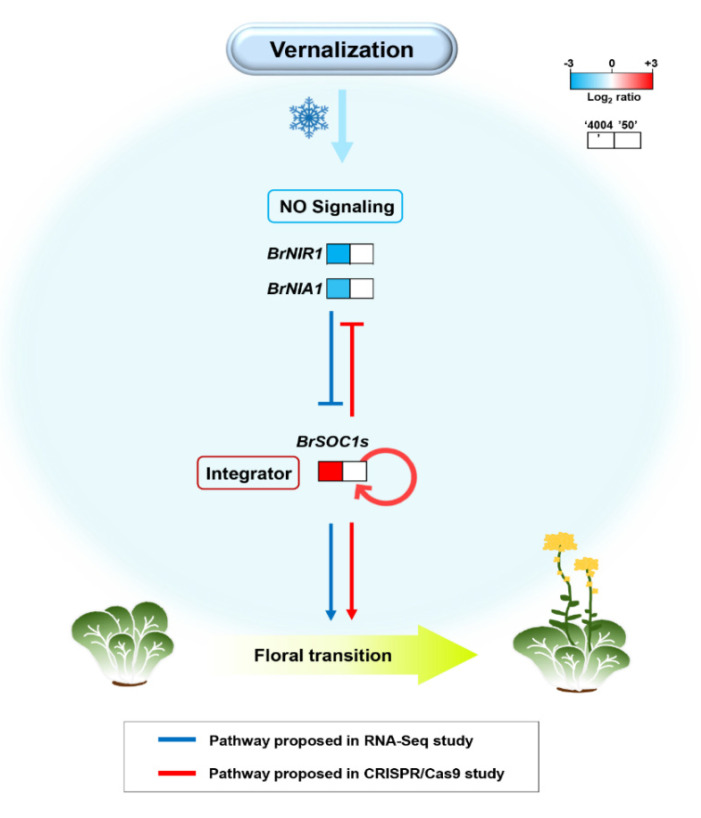
Schematic representation of the gene regulatory network controlling flowering time via interactions between nitrogen signaling and SOC1 response to vernalization in Chinese cabbage. The model is based on data obtained by qPCR and RNA-seq. The gene expression levels were normalized to the expression levels in line “50” after vernalization. Red indicates higher expression, and blue indicates lower expression relative to line “50” after 35 days of vernalization. Arrows indicate transcriptional activation, whereas bars indicate reciprocal transcriptional repressions. A red circular arrow indicates positive feedback regulation.

## Data Availability

All raw sequences were deposited in the GenBank Gene Expression Omnibus (GEO) under the accession numbers GSE106444 (for previous “4004” RNA-seq duplicates) and GSE139375.

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
