# Peer review of "Nitrogen Signaling Genes and SOC1 Determine the Flowering Time in a Reciprocal Negative Feedback Loop in Chinese Cabbage (Brassica rapa L.) Based on CRISPR/Cas9-Mediated Mutagenesis of Multiple BrSOC1 Homologs"

_ijms, 2021, doi:10.3390/ijms22094631_

Round 1

Reviewer 1 Report

The manuscript is well written and the data produced are conclusive and demonstrate the correlation of nitrogen signaling genes and flowering time. minor comments/edits before publication 

it would have be nice to try to increase the reads counts to have a higher number of DEGs. authors should address why their read counts was low 

line 396-397 sentence not clear

Author Response

Thank you very much for taking the time to provide a review of our manuscript.

  •  We explained the reasons why we used normalized read counts to identify flowering time-related DEGs between the two inbred lines in lines 268-272 (revised MS).
  • We have revised the sentence to clarify of the meaning (lines 402-404 in the revised MS)

Reviewer 2 Report

This manuscript examined the transcriptome profiles of two genotypes with different bolting time to reveal genes in the flowering pathway affected by vernalisation. It was presented thoroughly with sound scientific method. So I highly recommend to publish this paper. 

There are only few minor suggestions:

Introduction: there are a lot of information out there from not only Arabidopsis but also rice, cereal about flowering pathway and role of N in controlling flowering time. Please expand your introduction to cover those.

Line 201: lacking “of” after identification

Line 241: and also similar to rice system? As you indicated in line 237?

Line 279: Under the 0D condition again? Should be 35D?

Lin 318: “gibberellin and I” what does this mean?

Fig2: explanation for e) in Fig.2

Line 439-440: smaller font size?

481 and 482: yes, there are 4 main pathways, see Hill, C.B., and C. Li, 2016 Genetic Architecture of Flowering Phenology in Cereals and Opportunities for Crop Improvement. Frontiers in Plant Science 7 (1906)

483 and 494: so it is interesting but not new?

Line 503: why pick SOC but not CIR gene?

Author Response

Thank you very much for taking the time to review this manuscript and provide feedbacks.

Please find a point-by-point response to your comments.

  • We have now revised the manuscript to include the information on the N controlling flowering pathway in crop plants (lines 57-58 in the revised MS).
  • We have revised 2.4 heading title as your comment: 2.4 Identification of ~ (line 204).
  • We refered rice system for the explaning the molecular link between NO signaling genes and nitrogen content. Our results for nitrogen reductase and flowering relationship based on Arabidopsis system.
  • We have corrected the 0D Ver to 35D Ver (line 281 in the revised MS). 
  • We have revised I to Integrator as a full name (line 320 in the revised MS).
  • We have added Fig. 2e figure legend (lines 201-203 in the revised MS).
  • Unfortunately, I could not find a problem in the font size.
  • As shown Fig. 3a schematic daigram, we found a part of flowering mechanism in consistent with Arabidopsis in Chinese cabbage.
  • In order to develop late-bolting trait, we decided BrSOC1s to conduct genome editing in early bolting inbred line '20' because it is an essential activator in flowering pathway, but CIR is a negative regulator.

Reviewer 3 Report

The manuscript addresses an important topic of plant biology the flowering mechanisms of Brassica rapa, commonly known as  Chinese cabbage . The experimental plan includes two inbred lines, namely “4004” and “50”, that respond differently to bolting and flowering after the same vernalization conditions. The study presents a well-designed series of experiments utilizing cutting edge technology to dissect the differences in bolting and flowering time in  the two inbred lines.   

The authors used RNA-seq analyses to dissect the differential expression of genes implicated in bolting and flowering in both lines. They also Identified the implication of two genes of nitrogen metabolism, the nitrite reductase (NIR) and the nitrate reductase (NIA) in the differential induction of bolting in the two inbred lines. Using CRISPR/Cas9-mediated mutagenesis they verified the implication of the flowering enhancer genes BrSOC1s  in bolting and flowering in relation to the NIR and NIA gene expression.

The results are clearly presented; however, there are minor issues that should be elucidated more. These are the following:

  1. In the Introduction the genome size and the karyotype (2n) of the studied lines as well as of line “20” used in the transformation experiments  should be presented.
  2. Recent studies have shown that ploidy level in Brassica rapa may impact flowering development. Thus the authors are encouraged to address this issue in relation to the lines examined and the results.
  • A number of genes expression level was examined by RNA-seq to assess differences among the two lines . In the text (lines 300-330) most of the genes are shown by their abbreviated names. However, to gain a better understanding of the amount of work accomplished a Table showing the complete genes names and the role in flowering mechanism could be used. Alternatively a diagram presenting the role of these genes in the bolting-flowering  mechanism could be very helpful.

Author Response

Thank you very much for taking time to review this manuscript and provide feedbacks.

  • We have marked karyotype of used inbred lines for this study. The seed company (NongHyup Seed) generally develop diploid inbred lines (lines 99-105).
  • If you are kindly excuse me, I would like to avoid this commendation the reason is above.
  • We have revised the Table S5 for the full name of flowering time genes, as your comment.